# The peroxisomal exportomer directly inhibits phosphoactivation of the pexophagy receptor Atg36 to suppress pexophagy in yeast

**Houqing Yu[†], Roarke A Kamber[†‡], Vladimir Denic\***

Department of Molecular and Cellular Biology, Harvard University, Cambridge, United States

**\*For correspondence:**
vdenic@mcb.harvard.edu

[†]These authors contributed equally to this work

**Present address:** [‡]Department of Genetics, Stanford University, Palo Alto, United States

**Competing interest:** The authors declare that no competing interests exist.

**Abstract** Autophagy receptor (or adaptor) proteins facilitate lysosomal destruction of various organelles in response to cellular stress, including nutrient deprivation. To what extent membrane-resident autophagy receptors also respond to organelle-restricted cues to induce selective autophagy remains poorly understood. We find that latent activation of the yeast pexophagy receptor Atg36 by the casein kinase Hrr25 in rich media is repressed by the ATPase activity of Pex1/6, the catalytic subunits of the exportomer AAA+ transmembrane complex enabling protein import into peroxisomes. Quantitative proteomics of purified Pex3, an obligate Atg36 coreceptor, support a model in which the exportomer tail anchored to the peroxisome membrane represses Atg36 phosphorylation on Pex3 without assistance from additional membrane factors. Indeed, we reconstitute inhibition of Atg36 phosphorylation *in vitro* using soluble Pex1/6 and define an N-terminal unstructured region of Atg36 that enables regulation by binding to Pex1. Our findings uncover a mechanism by which a compartment-specific AAA+ complex mediating organelle biogenesis and protein quality control staves off induction of selective autophagy.

## Editor's evaluation

This study reveals how selective degradation of peroxisomes via autophagy (pexophagy) is repressed under unstressed conditions; the peroxisomal AAA+ ATPase complex binds to the pexophagy protein Atg36 to inhibit its phosphorylation by the casein kinase Hrr25, which triggers pexophagy. Thus, this work unveils a novel aspect in pexophagy regulation and provides mechanistic insights into the regulation of other selective autophagy pathways and other intracellular quality control systems.

## Introduction

The accumulation of damaged organelles is a hallmark of aging and neurodegenerative disease (*Mizushima and Levine, 2020*). In healthy cells, damaged organelles are detected and eliminated by selective autophagy, a process that involves the capture of organelles in transport vesicles called autophagosomes that deliver organelles to the lysosome for degradation (*Gatica et al., 2018*). A class of proteins called autophagy receptors targets a diverse range of organelles and other targets for autophagic degradation (*Kirkin and Rogov, 2019*). Recent work in yeast and mammals has established that autophagy receptors function not only by tethering autophagy targets to autophagosomes, but also by initiating *de novo* autophagosome formation via activation of the master autophagy kinase, Atg1/ULK1 (*Kamber et al., 2015*; *Ravenhill et al., 2019*; *Torggler et al., 2016*; *Vargas et al., 2019*).

To perform effective quality control, autophagy receptors must initiate degradation of damaged organelles while sparing functional organelles (*Green and Levine, 2014*). Several regulatory mechanisms that enable such selectivity by autophagy receptors have been identified. First, the localization of multiple autophagy receptors is regulated by organelle damage, such that some receptors localize only to damaged organelles (*Rogov et al., 2014*). Second, phosphorylation by cytosolic kinases is critical for the ability of several autophagy receptors to drive organelle degradation (*Farré et al., 2013*; *Tanaka et al., 2014*; *Wild et al., 2011*). Studies of mitochondrial autophagy in mammalian cells have shown that such receptor phosphorylation events can be regulated to enable effective organelle quality control: the kinase TBK1 is recruited to depolarized mitochondria, where it phosphoactivates the autophagy receptor OPTN, enabling selective elimination of damaged mitochondria (*Moore and Holzbaur, 2016*). For many other autophagy receptors that are known to be phosphorylated, however, it remains unclear whether their phosphorylation is regulated by signals of organelle damage in order to enable effective organelle quality control.

Recently, multiple groups working in yeast and mammalian cells reported intriguing observations that peroxisomes are rapidly degraded by autophagy in cells that lack Pex1, Pex6, or Pex15 (*Law et al., 2017*; *Nuttall et al., 2014*), which form a complex required for peroxisome matrix protein import (*Platta et al., 2013*). The Pex1/6 subunits form a hexameric AAA-ATPase that is recruited by Pex15 to peroxisomes, where it extracts ubiquitinated Pex5 (a receptor for peroxisome matrix proteins) from the peroxisome membrane to enable additional rounds of protein import (*Platta et al., 2005*). The accumulation of ubiquitinated proteins on the surface of peroxisomes has been shown to drive selective peroxisome autophagy (also known as pexophagy) in mammalian cells (*Deosaran et al., 2013*; *Kim et al., 2008*), and it was thus proposed that the accumulation of ubiquitinated Pex5 in exportomer-deficient cells explains why pexophagy is induced in those cells (*Law et al., 2017*; *Nazarko, 2017*). However, it remains to be determined why pexophagy is induced by exportomer mutations in yeast, as Pex5 is not required for pexophagy in exportomer-deficient cells and ubiquitination does not appear to be a pexophagy-inducing signal in yeast (*Nuttall et al., 2014*). All forms of budding yeast pexophagy, including pexophagy driven by exportomer loss, are dependent on the autophagy receptor Atg36 (*Motley et al., 2012*; *Nuttall et al., 2014*), whose ability to drive pexophagy depends on its phosphoactivation (*Farré et al., 2013*; *Tanaka et al., 2014*). Thus, to gain insight into the mechanism by which exportomer mutations induce peroxisome turnover by selective autophagy, we investigated the molecular basis of Atg36 regulation upon loss of exportomer subunits.

## Results

### Exportomer mutations enhance Atg36 phosphoactivation by Hrr25

We confirmed that deletion of the *PEX1* gene encoding a subunit of the exportomer promotes pexophagy under nitrogen starvation (*Nuttall et al., 2014*; *Figure 1—figure supplement 1A*). Furthermore, we confirmed that this process is dependent on the pexophagy receptor Atg36 as well as Atg11, a scaffold protein that connects autophagy receptors to the Atg1 complex involved in all forms of autophagy (*Figure 1—figure supplement 1A*), and is associated with a modified form of Atg36 that could be resolved by sodium dodecyl sulfate–polyacrylamide gel electrophoresis (SDS–PAGE; *Nuttall et al., 2014*; *Figure 1—figure supplement 1B*). Exportomer mutations have been previously shown to enhance removal of peroxisomes but not mitochondria or bulk cytoplasm, thus allowing the autophagy system to effectively exert a form of organelle quality control (*Nuttall et al., 2014*). Despite being selective, this pexophagy process resembles the mechanism of pexophagy induction by nitrogen starvation, which concomitantly elevates bulk autophagy and cytoplasm-to-vacuole (CVT) transport. In this latter, nonselective context, Atg36 is activated by Hrr25-mediated phosphorylation resulting in the enhanced ability of the modified receptor product to interact with Atg11 (*Tanaka et al., 2014*). We have obtained four lines of experimental evidence for an analogous mechanism by which exportomer mutants lead to Atg36 activation. First, lambda phosphatase treatment of Atg36-MYC isolated by affinity purification from a *pex1Δ* cell extract converted modified Atg36 to its unmodified form, thus rendering its migration indistinguishable from the wild-type control (*Figure 1A*). Second, Atg36 was no longer modified in the presence of a Pex3 mutation (*pex3-177*) (*Figure 1B*) that abolishes Atg36 phosphorylation by Hrr25 during nitrogen starvation (*Meguro et al., 2020*; *Motley et al., 2012*). Third, we took advantage of the previous observation that auxin-inducible degradation (AID)

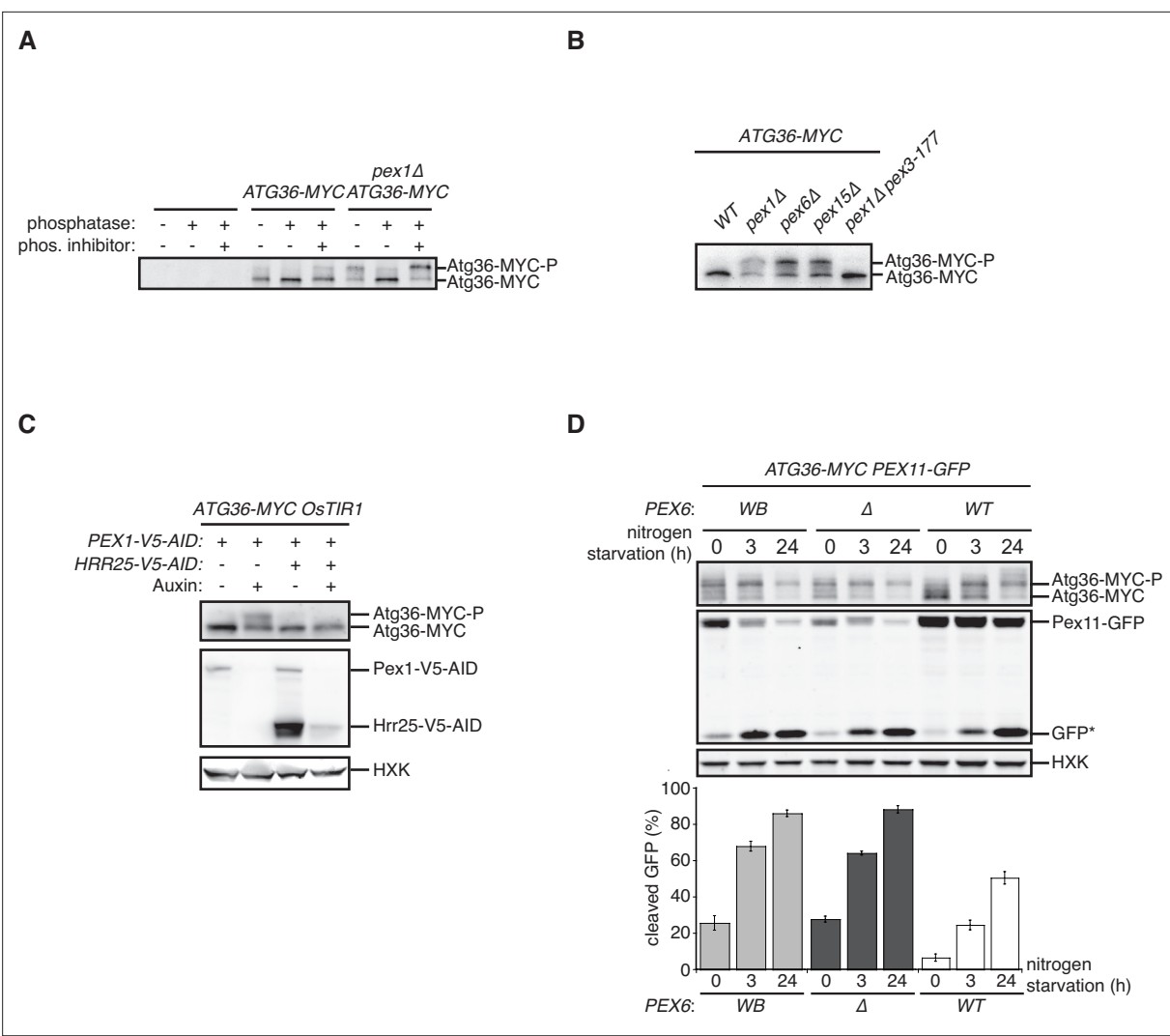

**Figure 1.** Exportomer mutations enhance Atg36 phosphoactivation by Hrr25. (**A**) The indicated extracts were subjected to immunoprecipitation (IP) with anti-MYC magnetic beads. After extensive washing, bound material was treated with lambda phosphatase in the presence or absence of phosphatase (phos.) inhibitors, and then resolved by sodium dodecyl sulfate–polyacrylamide gel electrophoresis (SDS–PAGE) followed by immunoblotting with anti-MYC antibodies. In previous studies, two Atg36 species were detected in the absence of nutrient stress or exportomer mutations (*Meguro et al., 2020*; *Nuttall et al., 2014*); the lower-mobility species might correspond to a basally phosphorylated form of Atg36. In our experiments with MYC-tagged Atg36, we were unable to detect more than one major Atg36 species in the absence of oleate stress or exportomer mutations. The higher- and lower-mobility species we detect under oleate stress or in the absence of exportomer function might correspond to basally phosphorylated and hyperphosphorylated species of Atg36, respectively. (**B**) Extracts derived from cells with indicated genotypes were resolved by SDS–PAGE followed by immunoblotting with anti-MYC antibodies. *pex3-177* indicates mutant version of Pex3 unable to bind Atg36 (*Motley et al., 2012*). (**C**) Extracts were prepared from cells with indicated genotypes that were treated for 3 hr with either 500 µM auxin or vehicle (DMSO) and resolved by SDS–PAGE, followed by immunoblotting with anti-MYC, anti-V5, and anti-HXK antibodies. (**D**) Cells expressing Pex11-GFP were grown in oleate medium for 22 hr and then transferred to nitrogen starvation medium. Extracts derived from cells with indicated genotypes were resolved by SDS–PAGE followed by immunoblotting with anti-MYC, anti-GFP, and anti-HXK antibodies. wt, wild-type allele of *PEX6*; Δ, genomic deletion of *PEX6*; WB, *PEX6* D2 Walker B motif mutant allele (E832A) at the endogenous *PEX6* locus. GFP*, GFP fragments produced upon vacuolar degradation of Pex11-GFP. HXK, hexokinase. Data points represent the mean values determined from three independent experiments. Error bars represent standard error.

The online version of this article includes the following source data and figure supplement(s) for figure 1:

**Source data 1.** Raw immunoblotting data related to *Figure 1*.

**Figure supplement 1.** The exportomer inhibits pexophagy in yeast.

of Pex1 can also enhance pexophagy (**Nuttall et al., 2014**; **Figure 1C**). We were then able to effectively suppress this conditional *pex1* phenotype by simultaneous AID of Hrr25 (**Figure 1C**). Lastly, we found that pexophagy enhanced by the loss of Pex1 was abolished by inhibiting the kinase activity of a Shokat mutant version of Hrr25 with the ATP analog 1NM-PP1 (**Figure 1—figure supplement 1A**). In sum, these data show that exportomer mutants enable Atg36 phosphoactivation by Hrr25, thus leading to selective pexophagy in the absence of nutritional stress.

The exportomer enables the recycling of several receptors for targeting and translocation of matrix proteins with peroxisome targeting signals (PTSs). This mechanoenzymatic process has not been fully reconstituted with purified components but appears to be driven by the Pex1/6 ATPase. To test if the latter activity also drives Atg36 repression, we mutated the D2 Walker B motif of Pex6, mutation of which was recently shown to abrogate ATP hydrolysis by the Pex1/6 complex *in vitro* , without affecting hexamerization of Pex6 with wild-type Pex1 subunits (**Ciniawsky et al., 2015**) or Pex6 binding to Pex15 (**Birschmann et al., 2003**). In cells expressing this point mutant version of Pex6 (E832A, Pex6[WB]), Atg36 phosphoactivation and pexophagy were induced to a similar degree as in *pex6Δ* cells (**Figure 1D**). These findings demonstrate that exportomer ATPase activity, either directly or indirectly, inhibits Atg36.

## Identification of physical interactions between Pex3 and the exportomer

PTS receptors have already been excluded as potential mediators of Atg36 activation in *pex1Δ* cells (**Nuttall et al., 2014**), but a systematic dissection of this mechanism *in vivo* had not been carried out. Thus, we devised an unbiased proteomics approach for detecting Pex1-dependent changes within the Atg36 interactome relevant to pexophagy. Atg36 has relatively low cell abundance and appears to be both inherently unstable and resistant to Hrr25 phosphorylation when not associated with Pex3, its peroxisomal anchor (**Meguro et al., 2020**). Thus, we chose Pex3-FLAG as our bait and subjected digitonin-solubilized extracts from postlogarithmic cells to affinity purification. Comparative mass spectrometry with extracts of control cells expressing untagged Pex3 revealed the presence of several previously identified Pex3 interaction partners, including Atg36, Inp1, Pex19, and Pex5, as well as several peroxisome membrane proteins whose insertion is known to be Pex19- and Pex3 dependent (**Fang et al., 2004**; **Hettema et al., 2000**; **Jones et al., 2001**; **Sacksteder et al., 2000**). Intriguingly, we also identified all three components of the exportomer, which to our knowledge had not been previously linked to Pex3. SDS–PAGE analysis followed by protein staining revealed two prominent high molecular weight species (>100 kDa) as likely candidates for Pex1 (~117 kDa) and Pex6 (~ 116 kDa) (**Figure 2A**). Indeed, we confirmed that Pex1 was a relatively abundant component of the Pex3 interactome by appending a bulky tag (3xV5-AID; ~30 kDa) to its C-terminus and detecting it by western blotting, as well as by the expected band shift following SDS–PAGE analysis (**Figure 2B**). We found that the coimmunoprecipitation between Pex3 and Pex1 was not affected by the expression of Pex6[WB] (**Figure 2C**), suggesting that the effect of this mutation on Atg36 phosphorylation and pexophagy induction (**Figure 1D**) is not simply caused by gross disruption of Pex3's association with the exportomer.

Next, we addressed the possibility that the exportomer inhibits Atg36 by remodeling the Pex3 interactome. Quantitative mass spectrometry analysis of immunopurified Pex3 from *pex1Δ* cells revealed the expected depletion of Pex6 (11-fold) and a lesser depletion of Pex15 (1.9-fold) (**Figure 2D**). By contrast, we observed no significant depletion or enrichment of Pex5, Pex19, or known Pex19 clients, suggesting that Pex1 loss does not alter the interaction between Pex3 and any of its other known binding partners detectable by this method (**Figure 2D**). In sum, our quantitative interactome analysis revealed a direct physical link between the exportomer and Atg36 bound to Pex3 but found no evidence of exportomer-dependent interactions that mediate Atg36 activation by Hrr25.

## Defining the mechanism by which the exportomer inhibits Atg36 *in vivo*

Our proteomics analysis raised the formal possibility Atg36 phosphoactivation in *pex1Δ* cells was enabled by Pex19 and its clients. However, we found that the absence of Pex19 was actually by itself sufficient to induce Atg36 phosphorylation comparably to exportomer *pex* mutations (**Figure 3A** and **Figure 1B**). Analysis of Pex3 immunopurified from *pex19Δ* cells revealed that the Pex3-Pex1

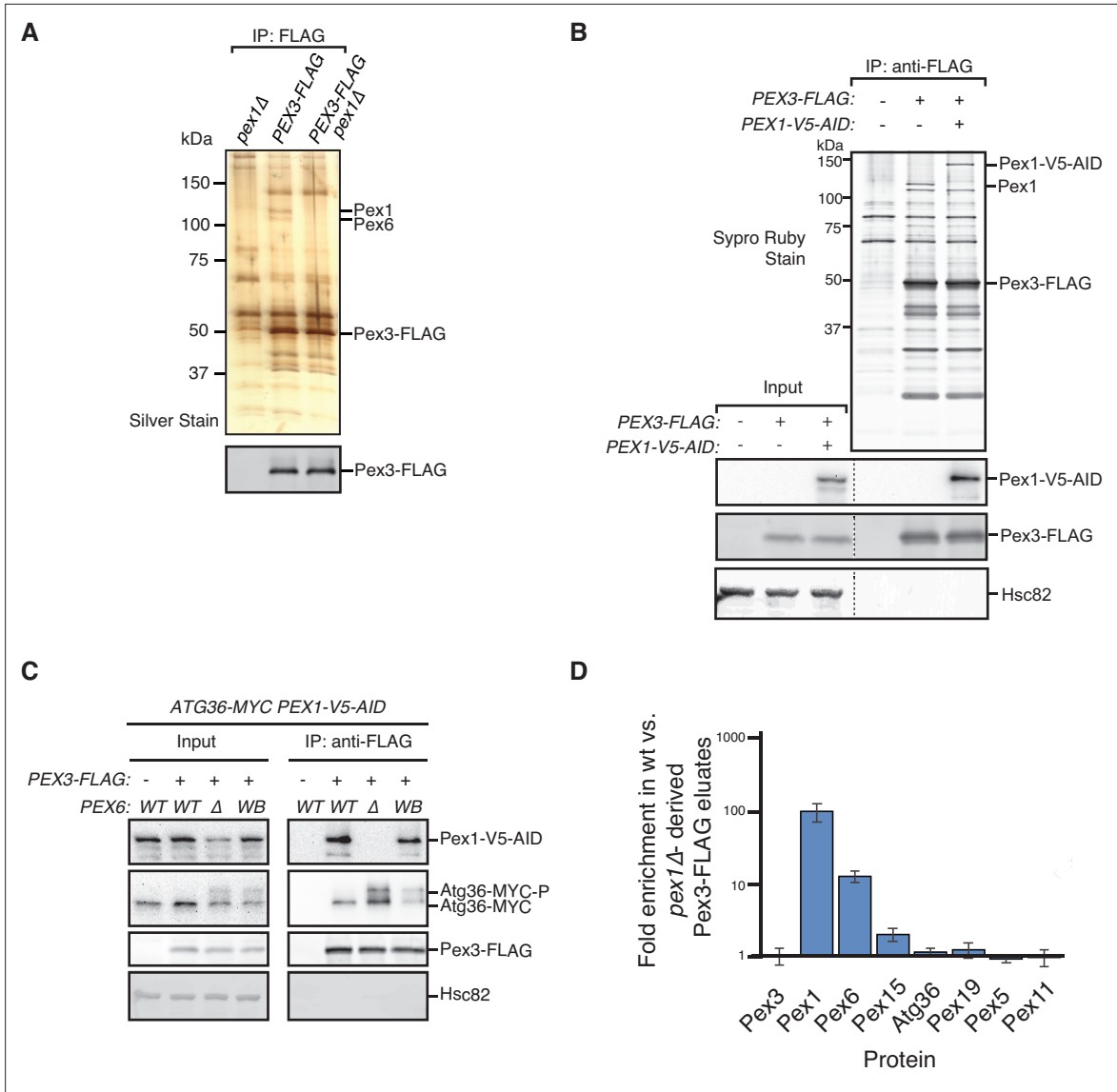

**Figure 2.** Identification of physical interactions between Pex3 and the exportomer. (**A, B**) Detergent-solubilized extracts derived from cells with indicated genotypes were first cleared at 100,000 × *g* and then immunoprecipitated (IP) with anti-FLAG magnetic beads. (**A**) Eluates were resolved by sodium dodecyl sulfate–polyacrylamide gel electrophoresis (SDS–PAGE) followed by either silver staining or immunoblotting with anti-FLAG antibodies. Selected proteins identified in the Pex3-FLAG eluate by mass spectrometry are indicated based on predicted molecular weight. (**B**) FLAG peptide eluates and extract (input) samples were resolved by SDS–PAGE followed by either SYPRO Ruby gel staining or immunoblotting with anti-V5, anti-FLAG, and anti-Hsc82 antibodies. Dotted line indicates splicing of identically processed gel-image data from the same gel. Hsc82 was used as sample processing control. (**C**) The indicated extracts were subjected to immunoprecipitation (IP) with anti-FLAG magnetic beads. FLAG peptide eluates and extract (input) samples were resolved by SDS–PAGE followed by immunoblotting with anti-V5, anti-MYC, anti-FLAG, and anti-Hsc82 antibodies. wt, wild-type allele of *PEX6*; Δ, genomic deletion of *PEX6*; WB, *PEX6* D2 Walker B motif mutant allele (E832A) at the endogenous *PEX6* locus. (**D**) Quantification of protein abundance in Pex3-FLAG eluates for selected proteins identified by mass spectrometry. Plotted is the mean ratio of protein abundance in wt-versus *pex1Δ*-derived Pex3-FLAG eluates. Error bars represent standard deviation.

The online version of this article includes the following source data for figure 2:

**Source data 1.** Raw staining, immunoblotting, and mass spectrometry data related to *Figure 2*.

**Source data 2.** The mass spectrometry data related to *Figure 2D*.

interaction was lost in *pex19Δ* cells, suggesting that the effect of Pex19 on Atg36 phosphorylation was indirectly caused by disruption of the exportomer's proximity to Atg36 (*Figure 3B*). To test this idea further, we devised an engineered system that bypasses Pex19's broad role in the assembly of peroxisomal membrane protein complexes. Our strategy was to first release Atg36 into the cytosol

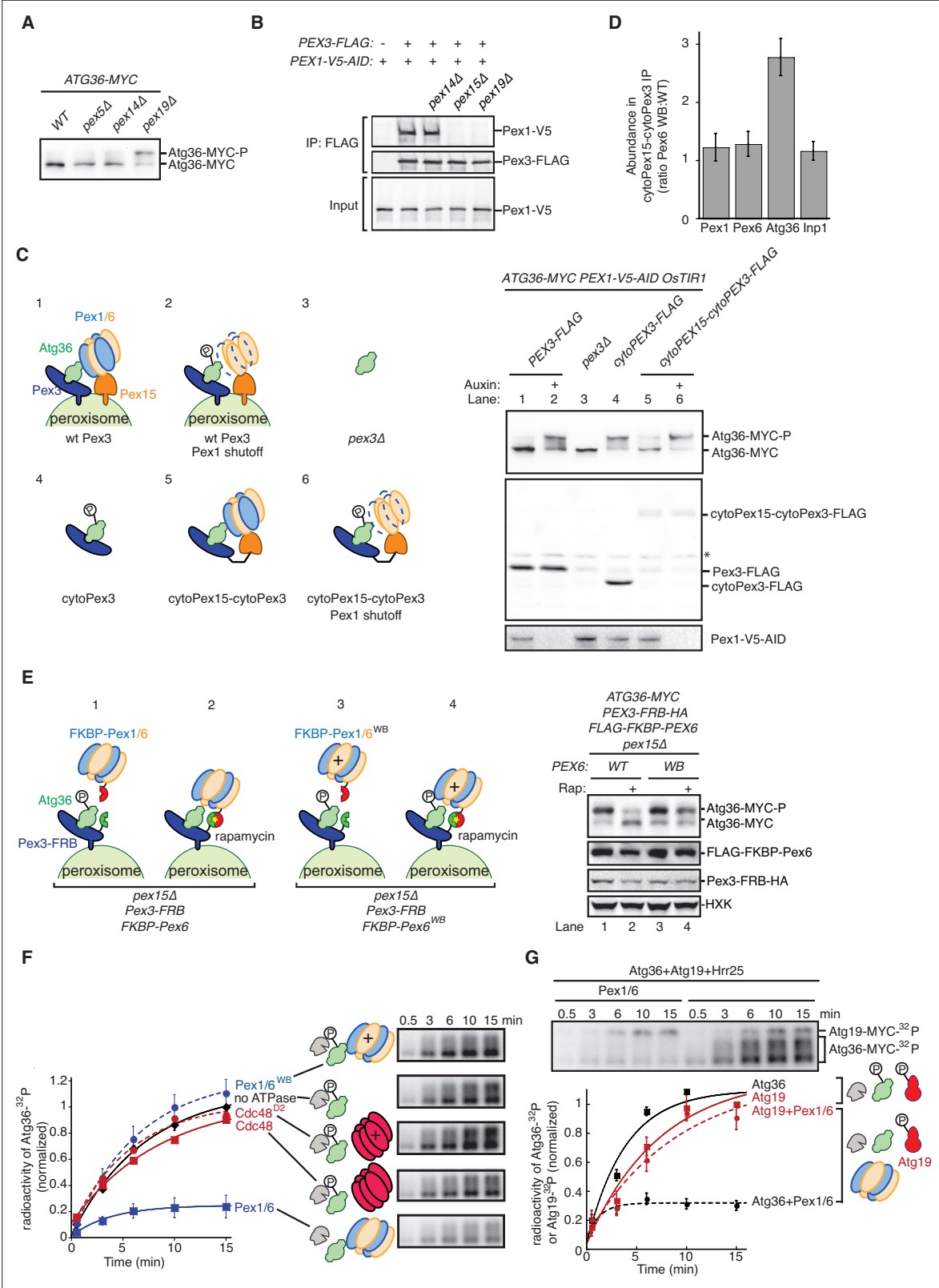

**Figure 3.** The exportomer inhibits Atg36 both *in vivo* and *in vitro*. (**A**) Extracts derived from cells with indicated genotypes were resolved by sodium dodecyl sulfate–polyacrylamide gel electrophoresis (SDS–PAGE) followed by immunoblotting with anti-MYC. wt, wild-type. (**B**) The indicated extracts were subjected to immunoprecipitation (IP) with anti-FLAG magnetic beads. FLAG peptide eluates and extract (input) samples were resolved by SDS–PAGE followed by immunoblotting with anti-FLAG and anti-V5 antibodies. (**C**) Left, schematic showing predicted pattern of Atg36 phosphorylation if

*Figure 3 continued on next page*

*Figure 3 continued*

exportomer is sufficient to inhibit Atg36. Right, extracts derived from cells with indicated genotypes that were treated for 3 hr with either 500 µM auxin or vehicle (DMSO) were resolved by SDS–PAGE followed by immunoblotting with anti-MYC, anti-FLAG, and anti-V5 antibodies, with lane numbers corresponding to numbered schematics. *, unspecific bands. (**D**) The abundance ratios of proteins in cytoPex15–cytoPex3-FLAG eluates derived from cells expressing *PEX6* D2 Walker B motif mutant allele at the endogenous *PEX6* locus (WB) versus eluates from cells expressing *PEX6* wild-type (wt). Error bars represent standard error. (**E**) Left, schematics showing the predicted pattern of Atg36 phosphorylation if bypassing Pex15 by an artificial tethering system (FRB-Rap-FKBP) is sufficient to inhibit Atg36 phosphorylation. Right, extracts derived from cells with indicated genotypes that were treated with 1 µM rapamycin (Rap) or vehicle (90% ethanol, 10% Tween-20) for 3 hr were resolved by SDS–PAGE followed by immunoblotting with anti-MYC, anti-FLAG, anti-HA, and anti-HXK antibodies, corresponding to schematics. WB, *PEX6* D2 Walker B motif mutant allele (E832A); HXK, hexokinase; Rap, rapamycin. (**F**) Purified Atg36 was incubated with $^{32}$P-labeled ATP, Hrr25, and indicated AAA-ATPases, and samples at each time point were resolved by SDS–PAGE followed by autoradiography to measure Atg36 phosphorylation. WB, Pex6 D2 Walker B motif mutant (E832A). D2, Cdc48 D2 ring mutant (E588Q). Data points represent the mean values determined from four repeat experiments. Error bars represent standard error. (**G**) Purified Atg36 and Atg19 were incubated together with $^{32}$P-labeled ATP, Hrr25, in the presence or absence of Pex1/6 complexes. Samples at each time point were resolved by SDS–PAGE followed by autoradiography. Data points represent the mean values determined from three repeat experiments. Note that the final time points for Atg19-only and Atg36-only conditions are superimposed. Error bars represent standard error.

The online version of this article includes the following source data and figure supplement(s) for figure 3:

**Source data 1.** Raw immunoblotting and mass spectrometry data related to *Figure 3*.

**Source data 2.** The mass spectrometry data related to *Figure 3D*.

**Figure supplement 1.** The exportomer inhibits Atg36 phosphorylation *in vivo*.

**Figure supplement 2.** The exportomer inhibits Atg36 phosphorylation *in vitro*.

by deleting the transmembrane domain TMD of Pex3 (cytoPex3, *Figure 3—figure supplement 1C, D*). A previous study focused on pexophagy regulation by nutritional stress showed that expressing a cytosolic form of Pex3 promotes Atg36 phosphorylation (*Meguro et al., 2020*). Similarly, we found that our cytoPex3 mutant also enabled Atg36 phosphorylation in the cytosol of postlogarithmic cells (*Figure 3C*) but was defective in Pex11-GFP processing (*Figure 3—figure supplement 1E*).

Next, we tested if recruitment of Pex1/6 was sufficient to inhibit Atg36 in this cytosolic context by constructing a chimeric protein scaffold comprising the cytosolic domain of Pex15 and the cytosolic domain of Pex3 fused with a short linker (cytoPex15–cytoPex3) (*Figure 3C*, *Figure 3—figure supplement 1C, D*). Strikingly, in cells expressing cytoPex15–cytoPex3, compared to cells expressing cytoPex3, we observed repression of Atg36 phosphorylation, but not when we caused AID of Pex1 (*Figure 3C*). Importantly, in this engineered context, repression was no longer dependent on Pex19 but remained driven by the exportomer's ATPase activity (*Figure 3—figure supplement 1A, B*). These data demonstrate that the exportomer's proximity to Atg36 is critical for inhibition of pexophagy but that Pex19 and peroxisomal membrane proteins are not strictly required for repression of Atg36 phosphorylation by Pex1/6.

To further define the core mechanism of Atg36 inhibition embedded within this stripped-down, membrane-free system, we examined the effect of the *pex6$^{WB}$* mutation on the composition of the cytoPex15–cytoPex3 complex. Quantitative proteomics analysis revealed that the amounts of Pex1 and Pex6 present in the cytoPex15–cytoPex3 complex were not affected, nor was the amount of Inp1, a soluble protein involved in peroxisome inheritance, which, like Atg36, binds Pex3 (*Figure 3D*; *Fagarasanu et al., 2005*, *Munck et al., 2009*). Notably, the abundance of Atg36 was increased ~2.2-fold in cytoPex15–cytoPex3 complexes isolated from cells expressing Pex6$^{WB}$ (*Figure 3D*). In sum, our interactome analysis supports the model that the exportomer directly blocks activation of Pex3-bound Atg36 by Hrr25.

A previous study focused on the exportomer's role in PTS receptor recycling noted the ability of Pex15's soluble domain to dynamically control the ATPase activity of Pex1/6 (*Gardner et al., 2018*). To determine if this is a general feature of the exportomer mechanism, we tested whether the requirement for Pex15 in Atg36 repression could be altogether bypassed by artificially tethering Pex6 to Pex3 using the rapamycin-dependent FRB/FKBP heterodimerization system (*Figure 3E*). In cells coexpressing Pex3-FRB and FKBP-Pex6, but not cells expressing Pex3-FRB and FKBP-Pex6$^{WB}$, addition of rapamycin repressed Atg36 phosphorylation (*Figure 3E* and *Figure 3—figure supplement 1F*). When a AAA ATPase involved in ER-associated degradation, Cdc48, was recruited to Pex3 through the similar FRB/Rap/FKBP system (cells coexpressing Pex3-FRB and FKBP-Cdc48), Atg36 phosphorylation was not repressed under rapamycin treatment (*Figure 3—figure supplement 1G*). Thus, Pex15

promotes Pex1/6 association with Pex3 on the surface of peroxisomes but is not strictly required for Atg36 inhibition.

## Pex1/6 ATPase inhibits Hrr25 phosphorylation of Atg36 *in vitro*

To further test whether Pex1/6 can directly suppress Atg36 phosphorylation by Hrr25 in the absence of additional factors, we first biochemically reconstituted the latter process using yeast-purified components (*Figure 3—figure supplement 2A*) and radiolabeled ATP. Incubation with wild-type but not kinase-dead (K38A) Hrr25 resulted in the time-dependent appearance of $^{32}$P-labeled Atg36 species with progressively reduced gel mobility following SDS–PAGE and autoradiography (*Figure 3—figure supplement 2B*). Strikingly, the addition of Pex1/6 purified from *E. coli* (*Figure 3—figure supplement 2A*) inhibited overall Atg36 phosphorylation by ~70%, whereas control incubations with Pex1/6$^{WB}$ were indistinguishable from those lacking Pex1/6 altogether (*Figure 3F* and *Figure 3—figure supplement 2C*).

We explored the specificity of the ability of Pex1/6 to repress Atg36 phosphorylation in this reconstituted system by analyzing additional reactions with factor substitutions. First, we replaced Pex1/6 with recombinant Cdc48 (and a control ATPase-defective mutant E558Q, Cdc48$^{D2}$; *Figure 3—figure supplement 2A*), a homomeric hexamer with a similar AAA+ architecture, and found only a marginal, ATPase-dependent effect on Atg36 phosphorylation (~10%). Notably, the wild-type preparations of Pex1/6 and Cdc48 had comparable basal ATPase activities (*Figure 3—figure supplement 2D*). Second, we substituted Atg19 for Atg36 as the substrate because this CVT pathway receptor also becomes phosphoactivated by Hrr25 during nutrient starvation (*Pfaffenwimmer et al., 2014*; *Tanaka et al., 2014*). This control reaction revealed that Pex1/6 did not inhibit Atg19 phosphorylation in the presence or absence of Atg36 (*Figure 3G*, *Figure 3—figure supplement 2E*). Thus, the Pex1/6 inhibition was specific to Atg36. Taken together with our previous *in vivo* evidence, these biochemical data provide complementary support for a model in which the exportomer directly inhibits Atg36 phosphoactivation by Hrr25.

## The exportomer inhibits pexophagy by binding to Atg36

Our *in vitro* evidence suggests that Atg36 is a novel exportomer substrate. AAA+ proteins can interact with their substrates using a variety of mechanisms but how the exportomer dislocates receptors (Pex5) for matrix proteins from the peroxisome membrane remains poorly understood (*Gardner et al., 2018*). Unlike these putative substrates, Atg36 is predicted to be a largely unstructured protein (*Figure 4A* and *Figure 4—figure supplement 1A*) and presented an opportunity to define the sequence features of this arguably simpler exportomer substrate. Thus, we took a yeast-two-hybrid (Y2H) approach and found that Atg36 selectively interacted with Pex1 but not Pex6 (*Figure 4B*). This interaction was fully abolished by N-terminal truncation of the first 30 amino acids of Atg36 (*Figure 4B*). This region is necessary but not sufficient for Atg36 interaction with Pex1, as a fragment corresponding to the N-terminal 59 amino acids of Atg36, but not a fragment corresponding to the N-terminal 30 amino acids, was able to interact with Pex1 (*Figure 4B*). Importantly, cell microscopy analysis revealed that even more severe N-terminal truncations of Atg36 still efficiently localized to peroxisomes (*Figure 4—figure supplement 1B*), thus pointing us to the existence of a C-terminal sequence determinant for peroxisome localization residing between amino acids 150 and 219 (*Figure 4—figure supplement 1B–D*).

A key prediction of the mechanistic model in which Pex1/6 represses pexophagy by binding to Atg36 is that mutations that selectively disrupt this interaction should lead to derepression. We found support for this idea by analyzing cells expressing N-terminal truncations of Atg36 following postlogarithmic growth (*Figure 4C* and *Figure 4—figure supplement 1E*). Specifically, Atg36 phosphorylation was enhanced by the *ΔN30atg36* mutation, which was also associated with an Atg11-dependent increase in Pex11-GFP processing (*Figure 4C* and *Figure 4—figure supplement 1F*). More severe truncations that are proximal to the Atg11-binding site resulted in the loss of this constitutive pexophagy phenotype (*Figure 4—figure supplement 1E*).

As the final test of the mechanism by which the exportomer inhibits Atg36, we used protein engineering to artificially restore the interaction between the *ΔN30atg36* mutant and the exportomer. Our strategy was to replace the missing Pex1-interacting region of Atg36 with a Pex6-interacting region of Pex15 (amino acids 1–30). As predicted, Pex15N-ΔN30Atg36 bound Pex6 as measured in a Y2H assay

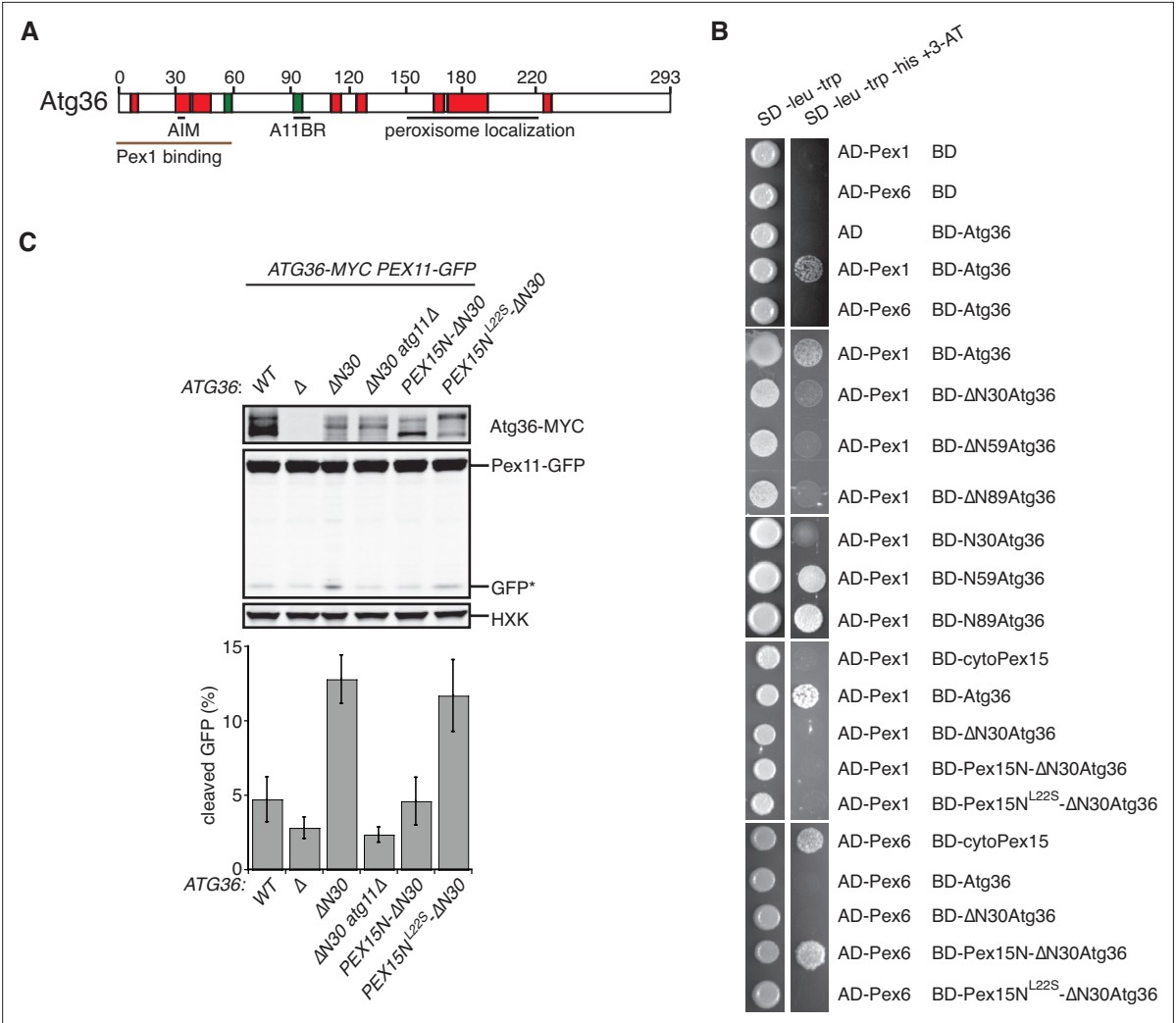

**Figure 4.** The exportomer inhibits pexophagy by binding to Atg36. (**A**) Schematic diagram of Atg36 with predicted secondary structural features indicated (PSIPRED 4.0) (*Jones, 1999*). Red region, predicted α-helix; green region, predicted β-sheet. AIM, Atg8-family-interacting motif, amino acids 33–36. A11BR, Atg11-binding region, amino acids 94–102. (**B**) Interaction between exportomer and Atg36 mutants in yeast-two-hybrid assay. Pex1 or Pex6 was fused with the Gal4 activation domain (AD) and Atg36 mutants were fused with Gal4 DNA-binding domain (BD). Cells expressing both AD-exportomer and BD-Atg36 mutants were grown on synthetic defined (SD) media plates as indicated. (**C**) Cells with indicated genotypes were grown in oleate medium for 22 hr. The extracts were resolved by sodium dodecyl sulfate–polyacrylamide gel electrophoresis (SDS–PAGE) followed by immunoblotting with anti-MYC, anti-GFP, and anti-HXK antibodies. GFP*, GFP fragments produced upon vacuolar degradation of Pex11-GFP. HXK, hexokinase. Data points represent the mean values determined from three independent experiments. Error bars represent standard error.

The online version of this article includes the following source data and figure supplement(s) for figure 4:

**Source data 1.** Raw immunoblotting data related to *Figure 4C*.

**Figure supplement 1.** The exportomer inhibits pexophagy by binding to Atg36.

(*Figure 4B*) and cells expressing Pex15N-ΔN30Atg36 exhibited both reduced Atg36 phosphorylation and reduced Pex11-GFP processing (*Figure 4C* and *Figure 4—figure supplement 1F*) relative to cells expressing *ΔN30atg36*. Importantly, both the Y2H interaction and pexophagy suppression phenotypes were abolished by introduction of a mutation in Pex15 that disrupts the Pex6–Pex15 interaction (L22S) (*Birschmann et al., 2003*; *Figure 4B, C*). In sum, these data further support a model in which the exportomer directly binds Atg36 to repress Atg36 phosphoactivation by Hrr25.

## Discussion

Yeast peroxisomes are constitutively degraded by selective autophagy in the absence of exportomer function (*Nuttall et al., 2014*), but the mechanistic basis for this phenomenon has remained unclear. In this work, we have presented evidence that the exportomer prevents pexophagy in unstarved cells by directly repressing phosphoactivation of the yeast pexophagy receptor, Atg36, by the casein kinase Hrr25. The ability of the exportomer to repress Atg36 phosphorylation is independent of its known role in membrane extraction of peroxisome proteins, as we showed that the exportomer is able to inhibit Atg36 phosphorylation away from the peroxisome or when reconstituted *in vitro* with purified soluble components. Repression of Atg36 phosphorylation by the exportomer requires its AAA-ATPase activity, but does not strictly require Pex15, whose function was bypassed by artificial tethering of Pex1/6 to Pex3.

We found that Pex1/6 physically interacts with Atg36 and mapped the binding site on Atg36 to its N-terminal region. Deletion of this region of Atg36 rendered it constitutively active and induced pexophagy, suggesting that the interaction between Atg36 and Pex1/6 is critical for repression of phosphorylation and pexophagy. However, the molecular mechanism by which Pex1/6 binding to Atg36 decreases phosphorylation of Atg36 by Hrr25 remains unclear. One possibility is that the N-terminal region of Atg36, which is near the critical Hrr25 S97 phosphosite for Atg11 binding (*Farré et al., 2013*), could provide a docking site for Hrr25, and that Pex1/6 binding to Atg36 disrupts the interaction between Hrr25 and Atg36. Future structural studies, for instance using hydrogen-deuterium exchange mass spectrometry to probe differences in Atg36 structure in the presence or absence of Pex1/6, should help resolve these issues.

Our work establishes that the ATPase activity of Pex1/6 can directly inhibit activation of Atg36 by Hrr25 but defining the mechanism of this mechanoenzymatic process remains an important future goal. One possibility is that active ATP hydrolysis enables Pex1/6 to interact more strongly with Atg36 and out-compete Hrr25 binding to Atg36. Alternatively, Pex1/6 may modulate the conformation of Atg36 and its abundance on the peroxisome similar to the activity of Pex1/6 in extracting Pex15 (*Gardner et al., 2018*). Additionally, it will be important to examine if growth and nutritional cues affect Atg36 phosphorylation via posttranslational modifications of Hrr25 and the exportomer.

Pex3 is required for Atg36 phosphorylation, and it was recently shown that Pex3 promotes Atg36 phosphoactivation by enhancing interaction of the Pex3–Atg36 complex with Hrr25 (*Meguro et al., 2020*). Due to technical difficulties in purifying a functional form of the soluble domain of Pex3, however, we have been unable to conduct quantitative enzymatic analyses of Pex3's role in promoting Hrr25 phosphorylation of Atg36, leaving uncertain whether the mechanism of stimulation involves substrate-induced conformational changes or a more passive tethering mechanism. It additionally remains unclear whether Pex3 regulates inhibition of Atg36 phosphorylation by Pex1/6, but our work suggests that Pex3 is not absolutely essential for this process, as Pex1/6 was capable of inhibiting Hrr25-mediated phosphorylation of Atg36 in *in vitro* assays conducted in the absence of Pex3. However, we cannot exclude that possible additional disruption of Atg36–Pex3 or Hrr25–Pex3 interactions by Pex1/6 may contribute to inhibition of Atg36 phosphorylation by the exportomer *in vivo*.

While Pex3 coimmunoprecipitates with exportmer subunits in this and previous studies (*Weir et al., 2017*), we were not able to distinguish whether the coimmunoprecipitation reflected direct interaction between Pex3 and Pex1/6/15 or was mediated by the unidentified factors. An early split-ubiquitin assay study reported a weak interaction signal between Pex15 and Pex3 (*Eckert and Johnsson, 2003*), but no binding between the cytosolic fragments of Pex3 (cytoPex3) and Pex15 (cytoPex15) in our Y2H assay was detected. We speculate that the transmembrane domains of Pex3 and Pex15 may mediate their putative interactions. Further structural and biochemical studies will help to test this hypothesis.

Atg36 is known to become phosphorylated by Hrr25 upon starvation, but future studies are required to determine how Atg36 phosphorylation is regulated by changes in nutrient status, and how starvation causes Atg36 to be released from, or overcome, inhibition by Pex1/6. It is known that starvation also increases Hrr25-mediated phosphorylation of two other autophagy receptors, Atg19 and Atg34 (*Mochida et al., 2014*; *Pfaffenwimmer et al., 2014*; *Tanaka et al., 2014*), so one possibility is that Hrr25 is globally activated by starvation signals, which may be sufficient to overcome Pex1/6 suppression. Alternatively, it is possible that exportmer activity (either toward all substrates or toward Atg36 specifically) decreases during starvation. In this vein, future studies should examine whether the sites at which Atg36 is phosphorylated by Hrr25 differ upon exportmer loss and upon

starvation, and whether exportomer loss synergizes with starvation to increase Atg36 phosphorylation and pexophagy.

Within the context of quality control, our findings raise the intriguing possibility that autophagy is capable of selective organelle clearance using a similar mechanistic logic to two protein homeostasis pathways: the heat shock response and the unfolded protein response (*Hetz et al., 2015*; *Hetz et al., 2020*; *Pincus, 2020*; *Richter et al., 2010*). A key feature of both is the dual ability of compartment-specific chaperones (cytosolic or endoplasmic reticulum [ER] resident, respectively) to enable negative feedback by functioning as both effectors of protein folding and degradation, as well as repressors of two conserved transcription factors (in yeast, Hsf1 and Ire1, respectively) that restore proteostasis by inducing gene expression of various proteostasis factors, including the chaperones themselves. By analogy, Pex1/6 are known to be mechanonenzymes for both membrane recycling and degradation of Pex5/7, the receptors for targeting soluble proteins to the peroxisome matrix (*Hagstrom et al., 2014*; *Kragt et al., 2005*; *Miyata and Fujiki, 2005*; *Platta et al., 2005*, *Platta et al., 2007*). We demonstrate that Pex1/6 also bind to Atg36 to directly repress wholesale peroxisome destruction but further work is needed to establish if physiological changes in the capacity of this AAA+ system are a readout of peroxisome dysfunction and damage. In this broad view of how different tiers of quality control can be logically connected, our work also raises the attractive possibility that emerging connections between ER protein folding stress and ER-phagy (*Molinari, 2021*) are the products of direct ER-phagy receptor regulation by ER-associated degradation factors.

Finally, even though the exportomer also appears to function as a repressor of selective pexophagy in mammalian cells (*Law et al., 2017*), it remains unclear whether the mechanism we have uncovered in yeast is relevant to the regulation of pexophagy in mammalian cells. Several important differences in the regulation of pexophagy between yeast and mammals – including the use of different autophagy receptor proteins and the apparently unique role of ubiquitin in mammalian pexophagy (*Deosaran et al., 2013*; *Kim et al., 2008*; *Nordgren et al., 2015*; *Zhang et al., 2015*) – warrant caution in seeking for convergence between these mechanisms. Systematic screening for potential kinase-mediated mechanisms involved in mammalian pexophagy should help resolve this issue. Regardless of these specifics, our work paves the way for future efforts to define a broadly conserved regulatory mechanism of selective autophagy by exploring the ability of organelle-localized enzymes to directly inhibit phosphoactivation of autophagy receptors.

# Materials and methods
## Contact for reagent and resource sharing

Further information and requests for resources should be directed to and will be fulfilled by the Lead Contact, Vladimir Denic (vdenic@mcb.harvard.edu).

## Yeast strain construction and PCRs

Yeast strains used in this study are listed in *Supplementary file 1*. Deletion strains were constructed in the BY4741 or W303 backgrounds (mating type **a**) by standard PCR-mediated gene knockout. *3 × FLAG, 13 × MYC, GFP, mCHERRY, NeonGreen, 3 × V5 AID, FRB-3 × HA*, and *3 × FLAG-FKBP12* (Euroscarf) cassettes were used to modify gene loci using standard PCR-mediated gene tagging. To introduce *OsTIR1* into the genome, a plasmid containing *OsTIR1* (*Nishimura et al., 2009*) provided by A. Amon was digested with *PmeI* and transformed into yeast for integration at the *leu2* or *his3* locus. The *as-HRR25* yeast strain bearing a gatekeeper mutation (I82G) in *HRR25* (*Petronczki et al., 2006*) at its endogenous locus was provided by C. Shoemaker. *cytoPEX3-FLAG* was constructed by replacing the endogenous *PEX3* ORF with an allele of *PEX3* lacking codons 2–45. *cytoPEX15–cytoPEX3-FLAG* was constructed by replacing the endogenous *PEX3* ORF with the product of an overlap extension PCR that fused, in frame, PCR-generated fragments corresponding to codons 1–330 of *PEX15*, a glycine–serine linker, and codons 46–441 of *PEX3*. The N-terminal truncated mutants of *ATG36* were constructed by replacing the endogenous *ATG36* ORF with the alleles of *ATG36* lacking the N-terminal region as indicated. The product of an overlap extension PCR that fused, in frame, two DNA fragments encoding 1–30 of *PEX15* and 31–294 of *ATG36* was used to generate *Pex15N-ΔN30Atg36* mutant. The cassettes expressing mCherry-labeled *PEX3* and mutants under *TDH3* promoter were integrated at the *trp1* locus.

Mutagenesis was performed using QuikChange (Stratagene) mutagenesis. Unless otherwise indicated, genomic allelic exchanges were performed using standard *URA3* replacement and 5-FOA counterselection. Primer sequences for all strain constructions are available upon request.

## Yeast cell growth and whole cell extract preparation for immunoblotting

Saturated, overnight cultures were diluted to 0.2 OD$_{600}$ units and grown for 5 hr in YPD media to mid-log phase unless indicated otherwise. Cells were collected by centrifugation at 2800 × g, stored at −80°C, resuspended in 100 µl SDS–PAGE sample loading buffer per OD$_{600}$ unit, and boiled for 5 min. Samples were cooled and cleared for 1 min at 15,000 × g before loading.

To induce Pex1 and Hrr25 degradation, cells were treated with 500 µM 3-indoleacetic acid (auxin) (Sigma) in DMSO or mock-treated with DMSO alone. To inhibit Hrr25(as), cells were treated with 100 µM 1NM-PP1 (Sigma) for 5 hr prior to sample collection. To induce the overexpression of target genes under a ZD promoter, cells were treated with 1 µM β-estradiol in ethanol or mock-treated with ethanol alone. In *Figure 3E* and *Figure 3—figure supplement 1F, G*, cells were treated with 1 µM rapamycin (LC Laboratories) in 90% ethanol/10% Tween-20 or mock-treated with 90% ethanol/10% Tween-20 alone.

## Yeast cell growth and lysate preparation for immunoprecipitations

Saturated, overnight cultures were diluted 1:100 and grown for 8 hr in YPD media (1% yeast extract, 2% peptone, 2% dextrose) to mid-log phase. YP5D (1% yeast extract, 2% peptone, 5% dextrose) media was seeded with logarithmically growing cells to achieve a final OD$_{600}$ of ~2.0–3.0 after 9–11 doublings. Cells were pelleted at 3000 × g for 20 min, washed with distilled water and pelleted in 50 ml Falcon tubes (3000 × g for 1 min). Washed cell pellets were weighed and resuspended in 1 ml lysis buffer (50 mM HEPES–KOH, pH 6.8, 150 mM KOAc, 2 mM MgCl$_2$, 1 mM CaCl$_2$, 0.2 M sorbitol) per 6 g pellet. Lysis buffer cell suspensions were frozen, dropwise, in liquid nitrogen and the resulting frozen material was ground in the presence of cOmplete protease inhibitor cocktail (Roche) using a Retsch PM100 ball mill (large scale, 1 l cultures) or Retsch MM400 ball mill (small scale, 25 OD$_{600}$ units maximum). Frozen lysate powder was stored at −80°C.

## Atg36-MYC immunoprecipitation and phosphatase treatment

After thawing 50 OD units frozen lysate powder in 0.5 ml HNP buffer (50 mM HEPES–KOH, pH 6.8, 150 mM KOAc, 2 mM MgOAc, 1 mM CaCl$_2$, 15% glycerol, 1% NP-40, 1× phosphatase inhibitors [5 mM sodium fluoride, 62.5 mM beta-glycerophosphate, 10 mM sodium vanadate, 50 mM sodium pyrophosphate]), lysates were cleared twice at 3000 × g at 4°C. Protein G Dynabeads (Invitrogen) were pre-equilibrated with mouse anti-MYC (9E10) antibody (Sigma) and added to clarified extract for 2 hr at 4°C. Beads were collected, washed four times in HN buffer (HNP buffer without phosphatase inhibitors), and resuspended in 40 µl lambda phosphatase buffer (NEB). Two separate 8 µl aliquots of beads derived from each strain were treated with 1 µl lambda phosphatase (NEB) in 10 µl reactions containing, additionally, either 1 µl 10× phosphatase inhibitors or 1 µl water for 30 min at room temperature, before the reaction was terminated with equal volume 2× SDS–PAGE sample loading buffer.

## Immunoprecipitation of Pex3-FLAG complexes

Frozen lysate powder was prepared as described above. After thawing 100 OD units frozen lysate powder in 1.6 ml HNP buffer, lysates were cleared twice at 3000 × g for 5 min, then at 100,000 × g for 30 min. Protein G Dynabeads were pre-equilibrated with mouse anti-FLAG antibody (Sigma) and added to clarified extract for 3 hr at 4°C. After extensive washing with HNP buffer, proteins were eluted via two sequential 30 min incubations with 20 µl 1 mg/ml 3× FLAG peptide in HNP, at room temperature. The two eluates were pooled and analyzed by immunoblotting and SYPRO Ruby staining. For cytoPex15–cytoPex3-FLAG and cytoPex3-FLAG purifications, immunoprecipitations were performed similarly, but HP buffer (i.e., lacking detergent) (50 mM HEPES–KOH, pH 6.8, 150 mM KOAc, 2 mM MgOAc, 1 mM CaCl$_2$, 15% glycerol, 1× phosphatase inhibitors) was used in place of HNP.

## Mass spectrometry

Affinity-purified complexes were prepared in triplicate (biological replicates), as described above, and analyzed by mass spectrometry at the Thermo Fisher Scientific Center for Multiplexed Proteomics

(Harvard Medical School) as described previously (*Kamber et al., 2015*). Briefly, FLAG peptide elutions were briefly resolved via SDS-PAGE, alkylated, digested with trypsin and labeled with Tandem Mass Tag 10-plex reagents, and analyzed by multiplexed quantitative mass spectrometry.

## Protein purification

### Pex1/6

The coding sequence of *Saccharomyces cerevisiae* Pex1 wild-type was fused with a C-terminal streptavidin-binding peptide tag and inserted into pCOLA-Duet vector. The expression plasmids of His-tagged Pex6 wild-type and mutants were gifts from A. Martin (*Gardner et al., 2018*). For both the wild-type and mutant complexes, Pex1 and Pex6 subunits were coexpressed in BL21-CodonPlus (DE3)-*RIPL* cells. The expression strain was grown in 2×YT media (16 g tryptone, 10 g yeast extract, and 5 g NaCl per liter) with ampicillin and kanamycin at 37°C and induced at an $OD_{600}$ of 0.6–0.9 with 0.4 mM IPTG for a further 16 hr growth at 18°C. The *E. coli* cells were harvested at 5000 × *g* for 10 min at 4°C and resuspended in Buffer A (25 mM HEPES, pH 7.6, 100 mM NaCl, 100 mM KCl, 10 mM $MgCl_2$, 0.5 mM EDTA, 20 mM imidazole, 0.5 mM ATP, 10% glycerol, and cOmplete protease inhibitor cocktail [Roche]) and were frozen at −80°C or lysed by homogenization (Avestin Emulsiflex C3) at 15,000 psi at 4°C. Cell debris and unlysed cells were pelleted at 48,000 × *g* and the supernatant was incubated with Ni-NTA resin (Qiagen) for 2 hr at 4°C. The resin was poured into a gravity flow column and washed with 100 ml Buffer A and 50 ml Buffer B (Buffer A plus 30 mM imidazole) sequentially. Pex1/6 complexes were eluted by 25 ml Buffer C (Buffer A plus 480 mM imidazole) and the eluate was incubated with 1 mL High Capacity Streptavidin Agarose resin (Pierce) for 2 hr at 4°C. The agarose resin was poured into a gravity flow column and washed with 50 ml Buffer A. Pex1/6 complexes were eluted by Buffer D (Buffer A plus 4 mM biotin). Purified Pex1/6 was dialyzed into Buffer A for storage and *in vitro* assays.

### Atg19

*S. cerevisiae* Atg19 was cloned into pGEX6P-1 resulting in a N-terminal GST fusion protein. The protein was purified from BL21-CodonPlus (DE3)-*RIPL* cells as described in the literature (*Sawa-Makarska et al., 2014*). Cells were grown at 37°C in LB media (10 g tryptone, 5 g yeast extract, and 10 g NaCl per liter) with ampicillin to an $OD_{600}$ of 0.6, induced with 0.4 mM IPTG for a further 16 hr growth at 18°C. Harvested cells were resuspended in Buffer E (50 mM HEPES, pH 7.5, 300 mM NaCl, 2 mM $MgCl_2$, 2 mM mercaptoethanol, and cOmplete protease inhibitor cocktail [Roche]) and were frozen at −80°C or lysed by homogenization (Avestin Emulsiflex C3) at 15,000 psi at 4°C. Cell debris and unlysed cells were pelleted at 48,000 × *g* and the supernatant was incubated with glutathione beads (GE Healthcare) for 1 hr at 4°C. The beads were poured into a gravity flow column and washed with 100 ml Buffer E. The protein was cleaved from the GST tag by incubation with PreScission protease (GE Healthcare) overnight at 4°C. The supernatant containing the protein was concentrated and applied to a Superdex 200 10/30 GL column (GE Healthcare) and eluted with Buffer F (25 mM HEPES, pH 7.5, 150 mM NaCl, and 1 mM DTT). Fractions containing pure protein were pooled, concentrated and stored at −80°C.

### Hrr25

Wild-type and kinase-dead mutant (K38A) versions of Hrr25 were purified as previously described (*Peng et al., 2015*). Cell strains carrying leucine-deficient plasmids of $P_{GAL}$-HRR25-Strep-6 × His and $P_{GAL}$-hrr25(K38A)-Strep-6 × His were gifts from D. Drubin (*Peng et al., 2015*). Cells were grown in 25 ml leucine-dropout medium containing 2% glucose at 30°C overnight. The culture were inoculated into 700 ml leucine-dropout medium containing 2% raffinose to $OD_{600}$ of ~2 at 30°C. 100 ml 20% galactose and 200 ml 5× YP (50 g yeast extract and 100 g peptone per liter) were added to final concentrations of 2% and 1× YP, respectively. After additional 6–7 hr growth at 30°C, cells were harvested, washed twice with ice-cold water, and resuspended at a ½ (vol/wt) ratio of lysis buffer/yeast, and frozen in drops in liquid $N_2$. The resulting frozen materials were lysed using a Retsch PM100 ball mill. Frozen lysate powder was stored at −80°C or thawed with 2× (wt/wt) ratio of Buffer G (25 mM HEPES, pH 7.6, 100 mM NaCl, 100 mM KCl, 10% glycerol, and cOmplete protease inhibitor cocktail [Roche]). Lysates were cleared twice at 3000 × *g* at 4°C to remove cell debris and unlysed cells. The supernatant was incubated with Ni-NTA resin (Qiagen) for affinity purification as described

above. The eluate from Ni-NTA resin was concentrated, applied to a Superdex 200 10/30 GL column (GE Healthcare) and eluted with Buffer H (25 mM HEPES, pH 7.6, 400 mM NaCl, 10% glycerol). Fractions containing pure protein were pooled, concentrated and stored at −80°C.

### Atg36

*S. cerevisiae* Atg36 fused with C-terminal myc tag was cloned into the plasmid vector from D. Drubin, resulting in a leucine-deficient plasmid: $P_{GAL}$-6 × His-Atg36−10 × myc. Atg36 was overexpressed in strain D1074 (*Mata lys2::Pgal1-GAL4 pep4::HIS3 bar1::hisG*) (**St-Pierre et al., 2009**) by galactose induction and cells were collected and lysed using a Retsch PM100 ball mill as described above. The frozen lysate powder was thawed with 2× (wt/wt) ratio of Buffer G and cleared twice at 1000 × *g* at 4°C to remove unlysed cells, and then at 20,000 × *g* at 4°C to collect the pellet. The pellet was dissolved in 10 ml Buffer G containing 6 M guanidinium chloride. The solution was further cleared at 20,000 × *g* at 4°C and the supernatant was incubated with Ni-NTA resin (Qiagen) for 2 hr at 4°C. The resin was poured into a gravity flow column and sequentially washed with Buffer G containing 6, 4, 2, 1, 0.6, 0.4, 0.2, 0.1, and 0.05 M guanidinium chloride; 100 ml Buffer G; and 50 ml Buffer G containing 50 mM imidazole. Atg36 was eluted from Ni-NTA resin by Buffer G containing 500 mM imidazole. The eluate was concentrated and applied to a Superdex 200 10/30 GL column (GE Healthcare) and eluted with Buffer G. Fractions containing pure protein were pooled, concentrated, and stored at −80°C.

Protein concentrations were determined by measuring light absorbance at 280 nm and using extinction coefficients predicted from the proteins' sequence (ExPASy's ProtParam). The integrity and purity of proteins were evaluated by SDS–PAGE. ATPase activity was measured as reported previously (**Nørby, 1988**).

### *In vitro* kinase assay

500 nM Atg36 was incubated at 30°C with wild-type Hrr25 or Hrr25 K38A mutant (200 nM in **Figure 3F** and **Figure 3—figure supplement 2C**, and 2 µM in **Figure 3—figure supplement 2B**), in the absence or presence of 200 nM ATPases (wild-type Pex1/6 and Walker B mutant [E832A], wild-type Cdc48 and D2 mutant [E588Q]) in kinase buffer (50 mM Tris–HCl, pH 7.5, 10 mM MgCl$_2$, 0.1 mM EDTA, 2 mM DTT, 1 mM ATP, 10 mM creatine phosphate, 0.2 mg/ml creatine kinase) supplied with γ-$^{32}$P-ATP (Perkin-Elmer) (**Figure 3F**). 1 µM Atg19 was incubated at 30°C with 500 nM wild-type Hrr25 in the absence or presence of 200 nM Pex1/6 (wild-type and Walker B mutant [E832A]) in kinase buffer (**Figure 3—figure supplement 2E**). 500 nM Atg36 and Atg19 was incubated together at 30°C with 200 nM wild-type Hrr25 in the absence or presence of 200 nM Pex1/6 (wild-type) in kinase buffer (**Figure 3G**). The samples at each time point were resolved by SDS–PAGE followed by autoradiography. The $^{32}$P-labeled species of Atg36 or Atg19 were quantified in FIJI.

### Pexophagy assay

Pexophagy was induced using culture conditions similar to those described previously (**Motley et al., 2012**; **Tanaka et al., 2014**). Yeast cultures grown in synthetic dextrose (SD) + casamino acids (CA) + ATU medium (0.17% yeast nitrogen base [YNB] without amino acids and ammonium sulfate [YNB w/o aa and as], 0.5% ammonium sulfate, 2% glucose, 0.5% CA, 0.002% adenine sulfate, 0.002% tryptophan, and 0.002% uracil) overnight were diluted 10-fold with SO + CA + ATU medium (0.17% YNB w/o aa and as, 0.5% ammonium sulfate, 0.12% oleate, 0.2% Tween 40, 0.1% glucose, 1% CA, 0.1% yeast extract, 0.002% adenine sulfate, 0.002% tryptophan, and 0.002% uracil) and grown for 20 hr. Cells were harvested, washed with autoclaved water twice and inoculated into SD-N medium (0.17% YNB w/o aa and as, 2% glucose). Samples at each time point were precipitated with 10% trichloroacetic acid and resolved by SDS–PAGE followed by western blot. The abundances of the target proteins were quantified in FIJI.

### Protein interaction analysis in Y2H system

The GAL4-based Matchmaker yeast 2-hybrid system (Clontech Laboratories Inc) was used for Y2H analysis. Full-length open reading frames or truncated forms of the target proteins were inserted into pGADT7 (AD) or pGBKT7 (BD) vectors (Clontech Laboratories). The AD and BD constructs were transformed into *S. cerevisiae* strains Y2H gold (Clontech Laboratories). The representative transformants on SD -Leu, -Trp plates were used for Y2H analysis. The strains were grown in SD -Leu -Trp medium

to log phase and plated on the selective plates (SD -Leu, -Trp or SD -Leu, -Trp, -His, with 10 mM 3-amino-1,2,4-triazole [3-AT]) to test protein interaction.

## Fluorescence microscopy

Cells were grown in synthetic dropout medium to logarithmic phase, concentrated, and imaged at room temperature using an oil-immersion ×63 objective (NA of 1.4) on a microscope (Axiovert 200 M; Carl Zeiss equipped with a Yokogawa CSU-10 spinning disk, or Zeiss LSM 980 Airyscan microscope). Images were acquired using a Cascade 512B EM-CCD detector (Photometrics) and the Metamorph 7.8.8 acquisition software (Molecular Devices) or Zen system (Zeiss). Images were converted to maximum intensity projections in FIJI, and adjusted for brightness and contrast – equally for all images within each panel – in Photoshop (Adobe) or FIJI.

## Acknowledgements

We thank T Rapoport for sharing his lab's Cdc48 preps and E O'Shea, A Amon, A Hansen, and A Martin for sharing additional reagents; S Ward, N Weir, C Shoemaker, and D Verhaagen for yeast strain construction; and members of the Denic lab for comments and scientific advice. We thank Douglas Richardson and Christian Hellriegel at Harvard Center for Biological Imaging for their help with fluorescence microscopy. We dedicate this paper to the memory of the late Peter Arvidson who had administratively supported our lab for many years. This work was supported by NIH R01 GM121419-01, 5R35GM127136 (to VD), and NSF GRFP DGE1144152 (to RAK).

## Additional information

### Funding

| Funder | Grant reference number | Author |
| --- | --- | --- |
| National Institutes of Health | R01 GM121419-01 | Vladimir Denic |
| National Institutes of Health | 5R35GM127136 | Vladimir Denic |
| National Science Foundation | GRFP DGE1144152 | Roarke A Kamber |

The funders had no role in study design, data collection, and interpretation, or the decision to submit the work for publication.

### Author contributions

Houqing Yu, Conceptualization, Data curation, Formal analysis, Investigation, Methodology, Resources, Validation, Visualization, Writing – original draft, Writing – review and editing; Roarke A Kamber, Conceptualization, Data curation, Formal analysis, Funding acquisition, Investigation, Methodology, Resources, Validation, Visualization, Writing – original draft, Writing – review and editing; Vladimir Denic, Conceptualization, Funding acquisition, Supervision, Writing – original draft, Writing – review and editing

### Author ORCIDs

Houqing Yu (ID) http://orcid.org/0000-0002-9196-8118
Roarke A Kamber (ID) http://orcid.org/0000-0001-8462-4782
Vladimir Denic (ID) http://orcid.org/0000-0002-1982-7281

### Decision letter and Author response

Decision letter https://doi.org/10.7554/eLife.74531.sa1
Author response https://doi.org/10.7554/eLife.74531.sa2

# Additional files

## Supplementary files
- Transparent reporting form
- Supplementary file 1. Table of yeast strains used in this study.
- Source data 1. Source data related to *Figures 1–4* and figure supplements.
- Source data 2. Raw immunoblot and autoradiography data, related to *Figures 1–4*.
- Source data 3. Raw immunoblot and autoradiography data, related to figure supplements.

## Data availability
All data generated or analysed during this study are included in the manuscript and supporting files.

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
