## [Editor Report]

This study reveals how selective degradation of peroxisomes via autophagy (pexophagy) is repressed under unstressed conditions; the peroxisomal AAA+ ATPase complex binds to the pexophagy protein Atg36 to inhibit its phosphorylation by the casein kinase Hrr25, which triggers pexophagy. Thus, this work unveils a novel aspect in pexophagy regulation and provides mechanistic insights into the regulation of other selective autophagy pathways and other intracellular quality control systems.

---

## [Decision Letter]

**Decision letter after peer review:**

Thank you for submitting your article "The peroxisomal exportomer directly inhibits phosphoactivation of the pexophagy receptor Atg36 to suppress pexophagy in yeast" for consideration by *eLife*. Your article has been reviewed by 3 peer reviewers, one of whom is a member of our Board of Reviewing Editors, and the evaluation has been overseen by David Ron as the Senior Editor. The following individual involved in review of your submission has agreed to reveal their identity: Prof. Suresh Subramani (Reviewer #3).

Essential revisions:

As you can see below, all reviewers positively evaluated your manuscript. However, we also agreed that there are still some important unanswered questions and some controls needed to justify the conclusions drawn. Among the reviewers' comments, we request you to address the following issues.

1) Testing the possibility that Pex1/6 acts against Hrr25 (Reviewer #2, comment 2).

2) Investigating the involvement of the autophagy initiation complex in pexophagy stimulation in the absence of Pex1/6-mediated repression (Reviewer #2, comment 3).

3) Adding a control in which Cdc48 is localized to Pex3 using the FRB-FKBP system

in the experiments to show the specificity of Pex1/Pex6 towards Atg36 (Reviewer #3, Major comments).

4) Confirming the localization of chimeras, cyto-Pex3 and cytoPex15-cytoPex3 (Reviewer #3, Major comments).

Meanwhile, we do not request experiments on how the negative regulation by Pex1/6 is turned off during pexophagy, which should be addressed in a future study (Reviewer #2, comment 1; Reviewer #3, the first paragraph in Major comments).

*Reviewer #1 (Recommendations for the authors):*

Autophagy is a degradation system that sequesters various intracellular materials within membrane vesicles called autophagosomes for their delivery to and degradation in lysosomes or vacuoles and thereby plays significant roles in the maintenance and regulation of a wide range of cellular functions. Since autophagic degradation often targets components critical for cell viability and growth, it must be tightly regulated. Indeed, previous studies have discovered regulatory mechanisms for different types of autophagy. In selective autophagy of peroxisomes (pexophagy) in the budding yeast *Saccharomyces cerevisiae*, previous studies have shown that phosphorylation of the pexophagy receptor Atg36 stimulates pexophagy by enhancing Atg36 activity to recruit core Atg proteins responsible for autophagosome formation. On the other hand, disrupting the peroxisomal AAA+ ATPase complex, which is composed of Pex1 and Pex6 and required for protein import into peroxisomes, was reported to somehow stimulate pexophagy even under conditions where pexophagy does not occur in wild-type cells, implying that this complex is involved in the repression of pexophagy under non-inducing conditions.

In this study, based on elegant protein engineering, quantitative mass spectrometry, and phosphorylation reactions reconstituted using purified proteins, the authors discovered the mechanism underlying this pexophagy repression. The N-terminal region of Atg36 is captured by Pex1, which, in a manner dependent on ATPase activity of the complex, prevents Atg36 from phosphorylation by Hrr25 on peroxisomes, resulting in pexophagy inhibition. This conclusion was strongly supported by a series of experiments, all of which were logically designed and well conducted. Details for interactions among proteins involved in this mechanism are still unclear, but key interactions were likely to be clarified in this study. An important remaining issue is how this repression mechanism is cancelled upon pexophagy stimulation (nitrogen starvation). It is also interesting to know the relationship between this newly-found inhibitory mechanism and the previously-described counteracting mechanism, Pex3-mediated stimulation of Atg36 phosphorylation by Hrr25 (the authors found in this study that the Pex1-Pex16 complex interacts with Pex3 via Pex15). The physiological impact of this mechanism also remains to be investigated. At any rate, this study unveils a novel aspect in pexophagy regulation and also provides mechanistic insights into the regulation of other selective autophagy pathways and other intracellular quality control systems.

As mentioned above, I find this study clearly shows a new mechanism by which pexophagy is repressed under non-inducing conditions. I raised major remaining issues but think that these should be addressed in future studies.

*Reviewer #2 (Recommendations for the authors):*

When pexophagy is induced in yeast, pexophagy receptor Atg36 is phosphorylated by Hrr25, and the phosphorylated Atg36 interacts with Atg11 to deliver peroxisome to the isolation membrane. However, it remains unclear how phosphorylation of Atg36 is suppressed to prevent unnecessary pexophagy induction at unstressed conditions. It was recently reported that pexophagy was promoted by the lack of peroxisomal AAA-ATPase Pex1/Pex6 or Pex15. Thus, the authors focused on Pex1/Pex6 and identified that Pex1/Pex6 interacts with Atg36 and suppresses the phosphorylation of Atg36. They also showed that repression of Atg36 phosphorylation by Pex1/Pex6 requires their AAA-ATPase activity, but not Pex15 function by using in vivo and in vitro assays,

Strengths:

By in vivo assays, the authors clearly showed that Pex1/Pex6 represses the phosphorylation of Atg36 by interacting with Atg36. The in vitro reconstitution assay support entirely the findings of in vivo that Pex1/Pex6 inhibits phosphorylation of Atg36 by Hrr25. The requirement of ATPase activity of Pex1/Pex6 for the repression of Atg36 phosphorylation was also demonstrated both in vivo and in vitro.

Using the transmembrane domain deleted Pex3 (cytoPex3), cytoPex15-cytoPex3 chimeric protein, and the rapamycin dependent FRB/FKBP system, the authors succeeded to show that inhibition of Atg36 phosphorylation works without Pex15 and peroxisomal membrane.

Weaknesses:

Although the authors found that Pex1/Pex6 suppresses phosphorylation of Atg36 and pexophagy, the mechanism by which Pex1/Pex6 suppresses phosphorylation of Atg36 is unclear. Furthermore, this study did not figure out how the phosphorylation of Atg36 is regulated by Pex1/Pex6, i.e., how Atg36 is phosphorylated by releasing the inhibition by Pex1/Pex6 when pexophagy should be induced.

1) It was shown that Pex1/Pex6 suppresses phosphorylation of Atg36. However, the mechanism which regulates phosphorylation of Atg36 is unclear. One possibility is that posttranslational modification such as phosphorylation or ubiquitination regulates the activity of Hrr25 or Pex1/Pex6. Another possibility is that the localization of Hrr25 or Pex1/Pex6 changes upon pexophagy induction. The authors should test these possibilities.

2) The AAA-ATPase activity of Pex1/Pex6 is required for the repression of Atg36 phosphorylation. It may be possible that the AAA-ATPase activity unfolds or degrades Hrr25 when Pex1 interacts with Atg36. This possibility can be tested by adding substrate of Casein Kinase 1 in in vitro reconstitution assay, as shown in Figure 3F.

3) In this study, pexophagy was induced at post-logarithmic growth in the oleate medium. In this culture condition, TOR may be suppressed and nonselective macroautophagy may be induced. Constitutive phosphorylation of Atg36 by Pex6WB or δ-N30-Atg36 can induce pexophagy. However, it is unclear whether phosphorylation of Atg36 alone can trigger pexophagy or activation of autophagy machinery is required in addition to Atg36 phosphorylation for pexophagy. To test these possibilities, test the pexophagy at early-logarithmic growth in glucose medium in the presence or absence of a constitutively active form of Atg13 (Kamada et al., Mol. Cell. Biol. 30, 1049-1058) in Pex6WB or δ-N30-Atg36 strains.

*Reviewer #3 (Recommendations for the authors):*

This paper by Yu et al., addresses the mechanism of how the yeast pexophagy receptor, Atg36, is kept in check, until needed, prior to its phosphoactivation by the casein kinase, Hrr25, during pexophagy in yeast. This works follows previous studies in Pichia pastoris and Saccharomyces showing that the peroxisomal membrane protein (PMP), Pex3, is necessary for the localization and phosphoactivation of Atg36 by Hrr25. This study now shows that Pex3, in addition to recruiting Atg36 to peroxisomes, also interacts with one component, Pex1, of the interacting AAA ATPases, Pex1 and Pex6, which block the phosphorylation of Atg36 by Hrr25. An ATPase-active Pex1/Pex6 interacts with Pex3 and inhibits Atg36 phosphorylation via interaction of Atg36 with Pex1, causing the exportomer to act as a negative regulator of pexophagy

Although previous work had shown the Pex1/Pex6 negatively regulate pexophagy, the mechanism was unknown, but is clarified here. The data are reasonably strong and convincing, but rely heavily on in vitro systems and artificial constructs, and could be strengthened by in vivo data and physiological constructs.

While the impact of the exportomer (the hexameric Pex1/Pex6 complex and the PMP, Pex15, which recruits Pex6 to peroxisomes) on Atg36 phosphoactivation and pexophagy is published, this study adds to the mechanistic details using several artificial constructs and in vitro reconstitution experiments. Because a good part of what is in the paper was shown previously (except for the new mechanistic details referenced above), what would be nice, but is missing, is how the ATPase activity of Pex1/Pex6 inhibits the phosphoactivation of Atg36.

In characterizing the Pex3 complexes, the authors make a new observation that all three components of the exportomer complex associate with Pex3, including a specific interaction with Pex1, along with other previously described partners. The authors go on to use engineered, chimeric protein constructs to reconstitute the interaction between Pex3, Atg36 and Pex1, as well as either the phosphorylation, or repression thereof, of Atg36, to show that the exportomer directly blocks activation of Pex3-bound Atg36 by Hrr25. However, the only caveat is the use of mislocalized and/or artificial chimeras, which should be strengthened by in vivo studies.

In showing the specificity of the Pex1/Pex6 ATPase towards blocking the phosphorylation of Atg36, the authors use another AAA ATPase, Cdc48, as a control (Figure 4F). This does not seem like a good control without localizing Cdc48 with Pex3 (using Pex3-FRB and FKBP-Cdc48 wt and mutant, +/- rapamycin as shown in Figure 3 E).

Are strains expressing various chimeras, such as cyto-Pex3, cytoPex15-cytoPex3 incompetent for physiological pexophagy and are these proteins really cytosolic as presumed, which may not be true because Pex3 associates with other PMPs? This is important to validate the claim that the exportomer's proximity to Atg36 is critical for inhibition of pexophagy.

---

## [Author Response]

Essential revisions:As you can see below, all reviewers positively evaluated your manuscript. However, we also agreed that there are still some important unanswered questions and some controls needed to justify the conclusions drawn. Among the reviewers' comments, we request you to address the following issues.1) Testing the possibility that Pex1/6 acts against Hrr25 (Reviewer #2, comment 2).

We thank the reviewer for this suggestion. To address the specificity of the inhibitory effect of Pex1/6 on Hrr25-mediated phosphorylation of Atg36, we analyzed the effect of Pex1/6 on the in vitro phosphorylation of Atg19, a receptor for a distinct form of selective autophagy and a *bona fide* Hrr25 substrate (Pfaffenwimmer et al., 2014; Tanaka et al., 2014). We previously showed that unlike Atg36, Atg19 phosphorylation is not subject to inhibition by Pex1/6 in vitro. To exclude the possibility that Hrr25 is globally compromised by the AAA+-substrate mechanics in the former reaction, we have now analyzed Atg19 and Atg36 phosphorylation by Hrr25 in the same reaction and found that these two substrate-kinase pairs behaved orthogonally in the presence of Pex1/6: Atg19 remained an equally good Hrr25 substrate while Atg36 inhibition persisted (new Figure 3G, Figure 3—figure supplement 2E). These data argue strongly against the possibility that Pex1 interaction with Atg36 leads to global Hrr25 kinase inhibition under our in vitro conditions.

2) Investigating the involvement of the autophagy initiation complex in pexophagy stimulation in the absence of Pex1/6-mediated repression (Reviewer #2, comment 3).

The reviewer has raised the interesting suggestion that by making Atg13 constitutively active we might be able to derepress pexophagy even in early-logarithmic conditions. Thus, we evaluated whether Pex6^WB^ or ∆N30Atg36 are able to induce pexophagy in cells grown in SD media to OD 0.5 and 1.0. We found that expression of Atg13-8SA as the sole constitutive copy of Atg13 did not result in elevated Pex11-GFP processing relative to cells with wild-type Atg13. Similarly, at the postlogarithmic phase (OD 5.2), Atg13 (8SA) only marginally increased Pex11-GFP processing and this effect appeared to be independent of the regulatory Atg36 phosphorylation mechanism defined by our study. Even though these new data suggest the existence of an undefined post-logarithmic signal they might also be explained by the insufficiency of Atg13-8SA as a strong inducer of nonselective macroautophagy. Because of these interpretation difficulties, we would prefer to leave them out of the manuscript and consider a more thorough investigation of this issue in the future.

**Author response image 1. sa2fig1:** 

3) Adding a control in which Cdc48 is localized to Pex3 using the FRB-FKBP systemin the experiments to show the specificity of Pex1/Pex6 towards Atg36 (Reviewer #3, Major comments).

We thank the reviewer for this suggestion and have engineered yeast strains for chemical heterodimerization of Cdc48 to Pex3 on the peroxisome membrane. To reduce the potential for disrupting Cdc48’s essential function in many cellular processes, we expressed a second copy of FLAG-FKBP-Cdc48 from a β-estradiol inducible promoter. Using a similar experimental setup to the one in which rapamycin enables FKBP-Pex6 to bind Pex3-FRB and repress Atg36 phosphorylation, we observed no repression by FKBP-Cdc48 expression even though the protein levels of FKBPCdc48 were substantially overexpressed relative to FKBP-Pex6 (Figure 3—figure supplement 1G). These new findings provide complementary in vivo support for the AAA+ specificity of Atg36 inhibition by Pex1/6 that we originally demonstrated solely by our in vitro kinase assay.

4) Confirming the localization of chimeras, cyto-Pex3 and cytoPex15-cytoPex3 (Reviewer #3, Major comments).

As the reviewer suggested, we analyzed the subcellular localization of the Pex3 chimeras lacking the native transmembrane domain. We expressed a second copy of either mCherry-tagged Pex3 or chimeras thereof in cells with a mutant allele of Pex3 (*pex3-177*) that is specifically defective in recruiting Atg36 and in pexophagy (Motley et al., 2012) but otherwise wild-type for peroxisome biogenesis. As expected, wt Pex3-mCherry formed puncta that colocalized with Pex11-GFP, while both cytoPex3-mCherry and cytoPex15-cytoPex3-mCherry had a diffuse cytoplasmic localization

(Figure 3—figure supplement 1D). We additionally confirmed that only wt Pex3-mCherry complemented the defect in pexophagy whereas the cytosolic Pex3 chimeras did not (Figure 3—figure supplement 1E).

Reviewer #2 (Recommendations for the authors):[…]1) It was shown that Pex1/Pex6 suppresses phosphorylation of Atg36. However, the mechanism which regulates phosphorylation of Atg36 is unclear. One possibility is that posttranslational modification such as phosphorylation or ubiquitination regulates the activity of Hrr25 or Pex1/Pex6. Another possibility is that the localization of Hrr25 or Pex1/Pex6 changes upon pexophagy induction. The authors should test these possibilities.

We thank the reviewer for raising these important points. The auto-phosphorylation of Hrr25 has been reported and observed in our in vitro kinase reactions. No ubiquitination or phosphorylation of Pex1/6 has been identified yet. We agree with the reviewer that there are posttranslational modifications, and other possible mechanisms regulating Atg36 phosphorylation that should be explored in the standalone study. To clarify the impact of the current work and more clearly frame the remaining mechanistic questions, we have described several competing mechanistic models in the Discussion section of the revised manuscript that we will discern between in follow-up studies.

2) The AAA-ATPase activity of Pex1/Pex6 is required for the repression of Atg36 phosphorylation. It may be possible that the AAA-ATPase activity unfolds or degrades Hrr25 when Pex1 interacts with Atg36. This possibility can be tested by adding substrate of Casein Kinase 1 in in vitro reconstitution assay, as shown in Figure 3F.

We thank the reviewer for this suggestion. To address the specificity of the inhibitory effect of Pex1/6 on Hrr25-mediated phosphorylation of Atg36, we analyzed the effect of Pex1/6 on the in vitro phosphorylation of Atg19, a receptor for a distinct form of selective autophagy and a *bona fide* Hrr25 substrate (Pfaffenwimmer et al., 2014; Tanaka et al., 2014). We previously showed that unlike Atg36, Atg19 phosphorylation is not subject to inhibition by Pex1/6 in vitro. To exclude the possibility that Hrr25 is globally compromised by the AAA+-substrate mechanics in the former reaction, we have now analyzed Atg19 and Atg36 phosphorylation by Hrr25 in the same reaction and found that these two substrate-kinase pairs behaved orthogonally in the presence of Pex1/6: Atg19 remained an equally good Hrr25 substrate while Atg36 inhibition persisted (new Figure 3G, Figure 3). These data argue strongly against the possibility that Pex1 interaction with Atg36 leads to global Hrr25 kinase inhibition under our in vitro conditions.

3) In this study, pexophagy was induced at post-logarithmic growth in the oleate medium. In this culture condition, TOR may be suppressed and nonselective macroautophagy may be induced. Constitutive phosphorylation of Atg36 by Pex6WB or δ-N30-Atg36 can induce pexophagy. However, it is unclear whether phosphorylation of Atg36 alone can trigger pexophagy or activation of autophagy machinery is required in addition to Atg36 phosphorylation for pexophagy. To test these possibilities, test the pexophagy at early-logarithmic growth in glucose medium in the presence or absence of a constitutively active form of Atg13 (Kamada et al., Mol. Cell. Biol. 30, 1049-1058) in Pex6WB or δ-N30-Atg36 strains.

The reviewer has raised the interesting suggestion that by making Atg13 constitutively active we might be able to derepress pexophagy even in early-logarithmic conditions. Thus, we evaluated whether Pex6^WB^ or ∆N30Atg36 are able to induce pexophagy in cells grown in SD media to OD 0.5 and 1.0. We found that expression of Atg13-8SA as the sole constitutive copy of Atg13 did not result in elevated Pex11-GFP processing relative to cells with wild-type Atg13. Similarly, at the postlogarithmic phase (OD 5.2), Atg13 (8SA) only marginally increased Pex11-GFP processing and this effect appeared to be independent of the regulatory Atg36 phosphorylation mechanism defined by our study. Even though these new data suggest the existence of an undefined post-logarithmic signal they might also be explained by the insufficiency of Atg13-8SA as a strong inducer of nonselective macroautophagy. Because of these interpretation difficulties, we would prefer to leave them out of the manuscript and consider a more thorough investigation of this issue in the future.

Reviewer #3 (Recommendations for the authors):This paper by Yu et al., addresses the mechanism of how the yeast pexophagy receptor, Atg36, is kept in check, until needed, prior to its phosphoactivation by the casein kinase, Hrr25, during pexophagy in yeast. This works follows previous studies in Pichia pastoris and Saccharomyces showing that the peroxisomal membrane protein (PMP), Pex3, is necessary for the localization and phosphoactivation of Atg36 by Hrr25. This study now shows that Pex3, in addition to recruiting Atg36 to peroxisomes, also interacts with one component, Pex1, of the interacting AAA ATPases, Pex1 and Pex6, which block the phosphorylation of Atg36 by Hrr25. An ATPase-active Pex1/Pex6 interacts with Pex3 and inhibits Atg36 phosphorylation via interaction of Atg36 with Pex1, causing the exportomer to act as a negative regulator of pexophagyAlthough previous work had shown the Pex1/Pex6 negatively regulate pexophagy, the mechanism was unknown, but is clarified here. The data are reasonably strong and convincing, but rely heavily on in vitro systems and artificial constructs, and could be strengthened by in vivo data and physiological constructs.While the impact of the exportomer (the hexameric Pex1/Pex6 complex and the PMP, Pex15, which recruits Pex6 to peroxisomes) on Atg36 phosphoactivation and pexophagy is published, this study adds to the mechanistic details using several artificial constructs and in vitro reconstitution experiments. Because a good part of what is in the paper was shown previously (except for the new mechanistic details referenced above), what would be nice, but is missing, is how the ATPase activity of Pex1/Pex6 inhibits the phosphoactivation of Atg36.

We thank the reviewer for pointing out this as a key remaining mechanistic question that is raised by our novel findings that Pex1/6 directly suppresses Atg36 phosphoactivation. While our new in vitro data (Figure 3G) further narrow down the mechanistic possibilities (i.e. by excluding the possibility that Pex1/6 causes global repression of Hrr25 kinase activity), we believe that fully resolving this question is better suited for a subsequent standalone study, likely relying on structural characterization of the interaction between Pex1/6 and Atg36. To clarify the impact of the current work and more clearly frame the remaining mechanistic questions, we have described several competing mechanistic models in the Discussion section of the revised manuscript that we will discern between in follow-up studies. One possibility is that an ATPase active conformation (probably the ADP bound state) of Pex1/6 is more competent for phospho-inhibition of Atg36, for instance because the ADP bound state of Pex1/6 may exhibit stronger binding affinity for Atg36 and may thus out-compete Hrr25 binding to Atg36. Alternatively, by analogy to Pex1/6’s proposed role in extracting Pex15 from the peroxisomes (Gardner et al., 2018), Pex1/6 may also regulate Atg36 by disrupting its interaction with the membrane and/or Pex3. Regardless of these mechanistically detailed issues, the current work demonstrates that the ATPase activity of Pex1/6 is critical for suppression of pexophagy through direct repression of Atg36 phosphoactivation. The ability of kinases in general and casein kinase in specific (e.g. Chino et al., 2022) to activate autophagy receptors is emerging as a new regulatory paradigm in the field. As such, we feel that our current work makes a concrete and significant mechanistic and conceptual advance by showing how a AAA+ machine can enable autophagy receptors to be phosphoregulated.

In characterizing the Pex3 complexes, the authors make a new observation that all three components of the exportomer complex associate with Pex3, including a specific interaction with Pex1, along with other previously described partners. The authors go on to use engineered, chimeric protein constructs to reconstitute the interaction between Pex3, Atg36 and Pex1, as well as either the phosphorylation, or repression thereof, of Atg36, to show that the exportomer directly blocks activation of Pex3-bound Atg36 by Hrr25. However, the only caveat is the use of mislocalized and/or artificial chimeras, which should be strengthened by in vivo studies.

We thank the reviewer for making this point here and below. As the reviewer suggested, we analyzed the subcellular localization of the Pex3 chimeras and found that by contrast with the peroxisomal localization of wt Pex3-mCherry, both cytoPex3-mCherry and cytoPex15-cytoPex3-mCherry were localized to the cytoplasm (Figure 3—figure supplement 1D). We additionally confirmed that cells expressing Pex3 chimeras were defective for physiological pexophagy, as measured in an assay for Pex11-GFP processing conducted under nitrogen starvation conditions (Figure 3—figure supplement 1E). A detailed description is included in the essential revision (#4) and below.

Our co-immunoprecipitation studies showed the exportomer complex (Pex1/6/15) associates with Pex3. However, we have not found any Y2H evidence for a direct interaction between Pex3 and any individual exportormer subunits. Thus, we speculate in the revised Discussion that Pex3 binding to Pex15 in the context of the native transmembrane complex enables it to associate with Pex1/6.

In showing the specificity of the Pex1/Pex6 ATPase towards blocking the phosphorylation of Atg36, the authors use another AAA ATPase, Cdc48, as a control (Figure 4F). This does not seem like a good control without localizing Cdc48 with Pex3 (using Pex3-FRB and FKBP-Cdc48 wt and mutant, +/- rapamycin as shown in Figure 3 E).

We thank the reviewer for this suggestion and have engineered yeast strains for chemical heterodimerization of Cdc48 to Pex3 on the peroxisome membrane. To reduce the potential for disrupting Cdc48’s essential function in many cellular processes, we expressed a second copy of FLAG-FKBP-Cdc48 from a β-estradiol inducible promoter. Using a similar experimental setup to the one in which rapamycin enables FKBP-Pex6 to bind Pex3-FRB and repress Atg36 phosphorylation, we observed no repression by FKBP-Cdc48 expression even though the protein levels of FKBPCdc48 were substantially overexpressed relative to FKBP-Pex6 (Figure 3—figure supplement 1G). These new findings provide complementary in vivo support for the AAA+ specificity of Atg36 inhibition by Pex1/6 that we originally demonstrated solely by our in vitro kinase assay.

Are strains expressing various chimeras, such as cyto-Pex3, cytoPex15-cytoPex3 incompetent for physiological pexophagy and are these proteins really cytosolic as presumed, which may not be true because Pex3 associates with other PMPs? This is important to validate the claim that the exportomer's proximity to Atg36 is critical for inhibition of pexophagy.

As the reviewer suggested, we analyzed the subcellular localization of the Pex3 chimeras lacking the native transmembrane domain. We expressed a second copy of either mCherry-tagged Pex3 or chimeras thereof in cells with a mutant allele of Pex3 (*pex3-177*) that is specifically defective in recruiting Atg36 and in pexophagy (Motley et al., 2012) but otherwise wild-type for peroxisome biogenesis. As expected, wt Pex3-mCherry formed puncta that colocalized with Pex11-GFP, while both cytoPex3-mCherry and cytoPex15-cytoPex3-mCherry had a diffuse cytoplasmic localization (Figure 3—figure supplement 1D). We additionally confirmed that only wt Pex3-mCherry complemented the defect in pexophagy whereas the cytosolic Pex3 chimeras did not (Figure 3—figure supplement 1E).

References

Birschmann I, Stroobants AK, Berg M van den, Schäfer A, Rosenkranz K, Kunau W-H, Tabak HF. 2003. Pex15p of *Saccharomyces cerevisiae* Provides a Molecular Basis for Recruitment of the AAA Peroxin Pex6p to Peroxisomal Membranes. Journal of Biological Chemistry 14:2226–2236. doi:10.1091/MBC.E02-11-0752

Chino H, Yamasaki A, Ode KL, Ueda HR, N 4 Noda N, Mizushima N. 2022. hosphorylation by casein kinase 2 ensures ER-phagy receptor TEX264 binding to ATG8 proteins. bioRxiv 4259:2022.02.11.480038. doi:10.1101/2022.02.11.480038

Gardner BM, Castanzo DT, Chowdhury S, Stjepanovic G, Stefely MS, Hurley JH, Lander GC, Martin A. 2018. The peroxisomal AAA-ATPase Pex1/Pex6 unfolds substrates by processive threading. Nature Communications 2018 9:1 9:1–15. doi:10.1038/s41467-017-02474-4

Hulmes GE, Hutchinson JD, Dahan N, Nuttall JM, Allwood EG, Ayscough KR, Hettema EH. 2020. The Pex3-Inp1 complex tethers yeast peroxisomes to the plasma membrane. The Journal of cell biology 219. doi:10.1083/JCB.201906021

Krikken AM, Wu H, de Boer R, Devos DP, Levine TP, van der Klei IJ. 2020. Peroxisome retention involves Inp1-dependent peroxisome-plasma membrane contact sites in yeast. The Journal of cell biology 219. doi:10.1083/JCB.201906023

Meguro S, Zhuang X, Kirisako H, Nakatogawa H. 2020. Pex3 confines pexophagy receptor activity of Atg36 to peroxisomes by regulating Hrr25-mediated phosphorylation and proteasomal degradation. Journal of Biological Chemistry 295:16292–16298. doi:10.1074/JBC.RA120.013565

Motley AM, Nuttall JM, Hettema EH. 2012. Pex3-anchored Atg36 tags peroxisomes for degradation in *Saccharomyces cerevisiae*. The EMBO Journal 31:2852–2868. doi:10.1038/EMBOJ.2012.151

Munck JM, Motley AM, Nuttall JM, Hettema EH. 2009. A dual function for Pex3p in peroxisome formation and inheritance. The Journal of cell biology 187:463–471. doi:10.1083/JCB.200906161

Pfaffenwimmer T, Reiter W, Brach T, Nogellova V, Papinski D, Schuschnig M, Abert C, Ammerer G, Martens S, Kraft C. 2014. Hrr25 kinase promotes selective autophagy by phosphorylating the cargo receptor Atg19. EMBO reports 15:862–870. doi:10.15252/EMBR.201438932

Rogov V, Dötsch V, Johansen T, Kirkin V. 2014. Interactions between Autophagy Receptors and Ubiquitin-like Proteins Form the Molecular Basis for Selective Autophagy. Molecular Cell 53:167– 178. doi:10.1016/J.MOLCEL.2013.12.014

Tanaka C, Tan L-J, Mochida K, Kirisako H, Koizumi M, Asai E, Sakoh-Nakatogawa M, Ohsumi Y, Nakatogawa H. 2014. Hrr25 triggers selective autophagy–related pathways by phosphorylating receptor proteins. Journal of Cell Biology 207:91–105. doi:10.1083/JCB.201402128

Weir NR, Kamber RA, Martenson JS, Denic V. 2017. The AAA protein Msp1 mediates clearance of excess tail-anchored proteins from the peroxisomal membrane. eLife 6. doi:10.7554/ELIFE.28507